# LINDA: Unsupervised Learning to Interpolate in Natural Language Processing

## Abstract

Despite the success of mixup in data augmentation, its applicability to natural language processing (NLP) tasks has been limited due to the discrete and variable-length nature of natural languages. Recent studies have thus relied on domain-specific heuristics and manually crafted resources, such as dictionaries, in order to apply mixup in NLP. In this paper, we instead propose an unsupervised learning approach to text interpolation for the purpose of data augmentation, to which we refer as "Learning to INterpolate for Data Augmentation" (**LINDA**), that does not require any heuristics nor manually crafted resources but learns to interpolate between any pair of natural language sentences over a natural language manifold. After empirically demonstrating the LINDA's interpolation capability, we show that LINDA indeed allows us to seamlessly apply mixup in NLP and leads to better generalization in text classification both in-domain and out-of-domain.

## 1 Introduction

Data augmentation has become one of the key tools in modern deep learning especially when dealing with low-resource tasks or when confronted with domain shift in the test time (Lim et al., 2019). Natural language processing (NLP) is not an exception to this trend, and since 2016 (Sennrich et al., 2016), a number of studies have demonstrated its effectiveness in a wide variety of problems, including machine translation and text classification (Ng et al., 2020; Kumar et al., 2020a). We can categorize data augmentation algorithms into two categories: (1) perturbing a single example and (2) interpolating a pair of examples.

An algorithm in the first category takes as input a single training example and produces (often stochastically) various perturbed versions of it. These perturbed examples are used together with the original target, to augment the training set. In the case of back-translation (Edunov et al., 2018), the target side of an individual training example is translated into the source language by a reverse translation model, to form a new training example with a back-translated source and the original target translation. A few approaches in this category have been proposed and studied in the context of NLP, including rule-based approaches (Wei & Zou, 2019b) and language model (LM) based approaches (Ng et al., 2020; Yi et al., 2021), with varying degrees of success due to the challenges arising from the lack of known metric in the text space.

An augmentation algorithm in the second category is often a variant of mixup Zhang et al. (2018); Verma et al. (2019); Zhao & Cho (2019) which takes as input a pair of training examples and (stochastically) produces an interpolated example. Such an algorithm often produces an interpolated target accordingly as well, encouraging a model that behaves linearly between any pair of training examples. Such an algorithm can only be applied when interpolation between two data points is well-defined, often implying that the input space is continuous and fixed-dimensional. Both of these properties are not satisfied in the case of NLP. In order to overcome this issue of the lack of interpolation in text, earlier studies have investigated rather ad-hoc ways to address them. For instance, Guo et al. (2019) pad one of two input sentences to make them of equal length before mixing a random subset of words from them to form an augmented sentence. (Yoon et al., 2021) goes one step beyond by using saliency information from a classifier to determine which words to mix in, although this relies on a strong assumption that saliency information from a classifier is already meaningful for out-of-domain (OOD) robustness or generalization. We found that these existing text-level

interpolation methods rely on hand-crafted rules. This motivates us to explore how to generate interpolated text over a natural language manifold without heuristic rules.

In this paper, we take a step back and ask what interpolation means in the space of text and whether we can train a neural net to produce an interpolated text given an arbitrary pair of text. We start by defining text interpolation as sampling from a conditional language model given two distinct text sequences. Under this conditional language model, both of the original sequences must be highly likely with the interpolated text, but the degree to which each is more likely than the other is determined by a given mixing ratio. This leads us naturally to a learning objective as well as a model to implement text interpolation, which we describe in detail in the main section §3. We refer to this approach of text interpolation as **L**earning to **IN**terpolate for **D**ata **A**ugmentation (**LINDA**).

We demonstrate the LINDA's capability of text interpolation by modifying and finetuning BART (Lewis et al., 2020) with the proposed learning objective on a moderate-sized corpus (Wikipedia). We manually inspect interpolated text and observe that LINDA is able to interpolate between any given pair of text with its strength controlled by the mixup ratio. We further use an automated metric, such as sentence similarity, and demonstrate that there is a monotonic trend between the mixup ratio and the similarity of generated interpolation to one of the provided text snippets. Based on these observations, we test LINDA in the context of data augmentation by using it as a drop-in interpolation engine in mixup for text classification. LINDA outperforms existing data augmentation methods in the in-domain and achieves a competitive performance in OOD settings. LINDA also shows its strength in low-resource training settings.

## 2 Background and Related Work

Before we describe our proposal on learning to interpolate, we explain and discuss some of the background materials, including mixup, its use in NLP and more broadly data augmentation in NLP.

### 2.1 Mixup

Mixup is an algorithm proposed by (Zhang et al., 2018) to improve both generalization and robustness of a deep neural net. The main principle behind mixup is that a robust classifier must behave smoothly between any two training examples. This is implemented by replacing the usual empirical risk, $\mathcal{R}_{\text{emp}} = \frac{1}{N} \sum_{n=1}^{N} l(x^n, y^n; \theta)$, with

$$\mathcal{R}_{\text{mixup}} = \frac{1}{N^2} \sum_{n,n'=1}^{N} \int_0^1 l(f_{\text{int}}^x(x^n, x^{n'}; \alpha),$$
$$f_{\text{int}}^y(y^n, y^{n'}; \alpha); \theta) \, \mathrm{d}\alpha, \tag{1}$$

where $x$ is a training example, $y$ is the corresponding label, $N$ is the total number of training examples, $\alpha$ is the mixup ratio, and $l(x, y; \theta)$ is a per-example loss. Unlike the usual empirical risk, mixup considers every pair of training examples, $(x^n, y^n)$ and $(x^{n'}, y^{n'})$, and their $\alpha$-interpolation using a pair of interpolation functions $f_{\text{int}}^x$ and $f_{\text{int}}^y$ for all possible $\alpha \in [0, 1]$.

When the dimensions of all input examples match and each dimension is continuous, it is a common practice to use linear interpolation.

### 2.2 Mixup in NLP

As discussed earlier, there are two challenges in applying mixup to NLP. First, data points, i.e. text, in NLP have varying lengths. Some sentences are longer than other sentences, and it is unclear how to interpolate two data points of different length. (Guo et al., 2019) avoids this issue by simply padding a shorter sequence to match the length of a longer one before applying mixup. We find this sub-optimal and hard-to-justify as it is unclear what a padding token means, both linguistically and computationally. Second, even if we are given a pair of same-length sequences, each token within these sequences is drawn from a finite set of discrete tokens without any intrinsic metric on them. (Chen et al., 2020) instead interpolates the contextual word embedding

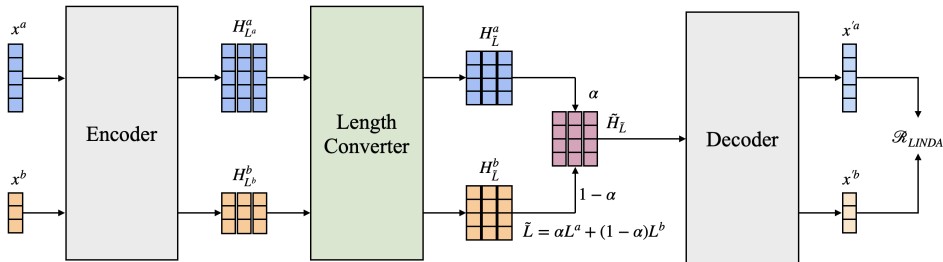

Figure 1: The overall architecture of our framework. Input texts $x^a$ and $x^b$ of which have possibly varying lengths are encoded and passed to the length converter to be transformed into representations with the interpolated matching length, $\tilde{L}$. $H^a_L$ and $H^b_L$, are interpolated into $\tilde{H}_L$ with the mixup ratio $\alpha$. The decoder is trained to reconstruct $\tilde{H}_L$ into $x^a$ and $x^b$ with the risk, $\mathcal{R}_{\text{LINDA}}$. We show an example where $x^a$ and $x^b$ has length 5 and 3, respectively and the hidden dimension size is 3.

vectors to circumvent this issue, although it is unclear whether token-level interpolation implies sentence-level interpolation. (Sun et al., 2020) on the other hand interpolates two examples at the sentence embedding space of a classifier being trained. This is more geared toward classification, but it is unclear what it means to interpolate two examples in a space that is not stationary (i.e., evolves as training continues.) (Yoon et al., 2021) proposes a more elaborate mixing strategy, but this strategy is heavily engineered manually, which makes it difficult to grasp why such a strategy is a good interpolation scheme.

## 2.3 Non-Mixup Data Augmentation in NLP

Although we focus on mixup in this paper, data augmentation has been studied before however focusing on using a single training example at a time. One family of such approaches is a token-level replacement. (Wei & Zou, 2019b) suggests four simple data augmentation techniques, such as synonym replacement, random insertion, random swap, and random deletion, to boost text classification performance. (Xie et al., 2020) substitutes the existing noise injection methods with data augmentation methods such as RandAugment (Cubuk et al., 2020) or back-translation. All these methods are limited in that first, they cannot consider larger context when replacing an individual word, and second, replacement rules must be manually devised.

Another family consists of algorithms that rely on using a language model to produce augmentation. (Wu et al., 2019) proposes conditional BERT contextual augmentation for labeled sentences using replacement-based methods. More recently, several works focus on large generative models to generate synthetic exampels for data augmentation (Anaby-Tavor et al., 2020; Kumar et al., 2020b; Yang et al., 2020). These augmentation strategies are highly specialized for a target classification task and often require finetuning with labeled examples. This makes them less suitable for low-resource settings in which there are by definition not enough labeled examples, to start with.

An existing approach that is perhaps closest to our proposal is self-supervised manifold-based data augmentation (SSMBA) by (Ng et al., 2020). SSMBA uses any masked language model as a denoising autoencoder which learns to perturb any given sequence along the data manifold. This enables SSMBA to produce realistic-looking augmented examples without resorting to any hand-crafted rules and to make highly non-trivial perturbation, beyond simple token replacement. It also does not require any labeled example.

(Ng et al., 2020) notice themselves such augmentation is nevertheless highly local. This implies that an alternative approach that makes non-local augmentation may result in a classifier that behaves better over the data manifold and achieves superior generalization and robustness. This is the goal of our proposal which we describe in the next section.

## 3 LINDA: Learning to Interpolate for Data Augmentation

In this section, we first define text interpolation (discrete sequence interpolation) and discuss desiderata that should be met. We then describe how we implement this notion of text interpolation using neural sequence modeling. We present a neural net architecture and a learning objective function used to train this neural network and show how to use a trained neural network for data augmentation.

### 3.1 Text Interpolation

We are given two sequences (text), $x^a = (x^a_1, \ldots, x^a_{L^a})$ and $x^b = (x^b_1, \ldots, x^b_{L^b})$.[1] They are potentially of two different lengths, i.e., $L^a \neq L^b$, and each token is from a finite vocabulary, i.e., $x^a_i \in \Sigma$ and $x^b_j \in \Sigma$. With these two sequences, we define text interpolation as a procedure of drawing another sequence from the following conditional distribution:

$$p(y|x^a, x^b, \alpha) = \prod_{t=1}^{L^y} p(y_t|y_{<t}, x^a, x^b, \alpha), \tag{2}$$

where $\alpha \in [0, 1]$ is a mixup ratio, and $L^y$ is the length of an interpolated text $y$.

In order for sampling from this distribution to be *text interpolation*, four conditions must be met. Let us describe them informally first. First, when $\alpha$ is closer to 0, a sample drawn from this distribution should be *more* similar to $x^a$. When $\alpha$ is closer to 1, it should be *more* similar to $x^b$. It is however important that any sample must be similar to *both* $x^a$ and $x^b$. Finally, this distribution must be smooth in that we should be able to draw samples that are neither $x^a$ and $x^b$. If the final two conditions were not met, we would not call it "inter"polation.

Slightly more formally, these conditions translate to the following statements:

**C1.** $\frac{p(x^a|x^a,x^b,\alpha)}{p(x^b|x^a,x^b,\alpha)} > 1$ if $\alpha < 0.5$.

**C2.** $\frac{p(x^a|x^a,x^b,\alpha)}{p(x^b|x^a,x^b,\alpha)} < 1$ if $\alpha > 0.5$.

**C3.** $p(x^a|x^a, x^b, \alpha) > 0 \wedge p(x^b|x^a, x^b, \alpha) > 0$

**C4.** C4. $p(x^a|x^a, x^b, \alpha) + p(x^b|x^a, x^b, \alpha) \ll 1$.

In the rest of this section, we describe how to parametrize this interpolation distribution with a deep neural network and train it to satisfy these conditions, for us to build the first learning-based text interpolation approach.

### 3.2 Parametrization

As shown in Figure 1, LINDA uses an encoder-decoder architecture to parametrize the interpolation distribution in Eq. equation 2, resembling sequence-to-sequence learning (Sutskever et al., 2014; Cho et al., 2014). The encoder takes the inputs respectively and delivers the encoded examples to the length converter, where each encoded example is converted to the same length. Matching length representations of the inputs are then interpolated with $\alpha$. The decoder takes the interpolated representation and reconstructs the original inputs $x^a$ and $x^b$ proportionally according to the given $\alpha$ value.

**Encoder**   We assume the input is a sequence of discrete tokens, $x = (x_1, \ldots x_L) \in \Sigma^L$, where $\Sigma$ is a finite set of unique tokens and $L$ may vary from one example to another. The encoder, which can be implemented as a bidirectional recurrent network (Bahdanau et al., 2015) or as a transformer (Vaswani et al., 2017), reads each of the input pairs and outputs a set of vector representations; $H^a = \{h^a_1, \ldots, h^a_{L^a}\}$ and $H^b = \{h^b_1, \ldots, h^b_{L^b}\}$.

---

[1]We use text and sequence interchangeably throughout the paper, as both of them refer to the same thing in our context.

**Length Converter**   Unlike in images where inputs are downsampled to a matching dimension, the original lengths of the input sequences are preserved throughout the computation in NLP. To match the varying lengths, preserving the original length information, we adapt a location-based attention method proposed by (Shu et al., 2020). First, we set the target length, $\tilde{L}$, as the interpolated value of $L^a$ and $L^b$: $\tilde{L} = \lceil \alpha L^a + (1-\alpha)L^b \rceil$.

Each vector set is either down- or up-sampled to match the interpolated length $\tilde{L}$ which results in $\tilde{H}^a = \{\tilde{h}_1^a, \ldots, \tilde{h}_{\tilde{L}}^a\}$ and $\tilde{H}^b = \{\tilde{h}_1^b, \ldots, \tilde{h}_{\tilde{L}}^b\}$. Each $\tilde{h}$ is a weighted sum of the hidden vectors: $\tilde{h}_j = \sum_{k=1}^{L} w_k^j h_k$ with $w_k^j = e^{a_k^j} / \sum_{k'} e^{a_{k'}^j}$ and $a_k^j = -\frac{1}{2\sigma^2}(k - \frac{L}{\tilde{L}}j)^2$ . $\sigma$ is a trainable parameter. We finally compute the interpolated hidden vector set $\tilde{H} = \{\tilde{h}_1, \ldots, \tilde{h}_{\tilde{L}}\}$ by

$$\tilde{h}_i = \alpha\tilde{h}_i^a + (1-\alpha)\tilde{h}_i^b. \tag{3}$$

**Decoder**   The decoder, which can be also implemented as a recurrent network or a transformer with causal attention, takes $\tilde{H}$ as an input and models the conditional probability distribution of the original input sequences $x^a$ and $x^b$ accordingly to the mixup ratio $\alpha$. The decoder's output is then the interpolation distribution $p(y|x^a, x^b, \alpha)$ in Eq. equation 2.

### 3.3   A Training Objective Function

We design a training objective and regularization strategy so that a trained model is encouraged to satisfy the four conditions laid out above and produces a text interpolation distribution.

**A main objective.**   We design a training objective function to impose the conditions laid out earlier in §3.1. The training objective function for LINDA is

$$\mathcal{R}_{\text{LINDA}} = -\frac{1}{N^2} \sum_{n,n'=1}^{N} \int_0^1 \alpha \log(p(x^n|x^n, x^{n'}, \alpha)$$
$$+ (1-\alpha)\log(p(x^{n'}|x^n, x^{n'}, \alpha))\mathrm{d}\alpha, \tag{4}$$

where we assume there are $N$ training examples and consider all possible $\alpha$ values equally likely. Minimizing this objective function encourages LINDA to satisfy the first three conditions; a trained model will put a higher probability to one of the text pairs according to $\alpha$ and will refrain from putting any high probability to other sequences.

Because $\mathcal{R}_{\text{LINDA}}$ is computationally expensive, we resort to stochastic optimization in practice using a small minibatch of $M$ randomly-paired examples at a time with $\alpha \sim \mathcal{U}(0,1)$:

$$\tilde{\mathcal{R}}_{\text{LINDA}} = -\frac{1}{M} \sum_{m=1}^{M} \alpha \log p(x^{a,m}|x^{a,m}, x^{b,m}, \alpha^m)$$
$$+ (1-\alpha)\log p(x^{b,m}|x^{a,m}, x^{b,m}, \alpha^m). \tag{5}$$

It is however not enough to minimize this objective function alone, as it does not prevent some of the degenerate solutions that violate the remaining conditions. In particular, the model may encode each possible input as a set of unique vectors such that the decoder is able to decode out both inputs $x^a$ and $x^b$ perfectly from the sum of these unique vector sets. When this happens, there is no meaningful interpolation between $x^a$ and $x^b$ learned by the model. That is, the model will put the entire probability mass divided into $x^a$ and $x^b$ but nothing else, violating the fourth condition. When this happens, linear interpolation of hidden states is meaningless and will not lead to meaningful interpolation in the text space, violating the third condition.

**Regularization**   In order to avoid the degenerate solution, we introduce three regularization techniques. First, inspired by the masked language modeling suggested in Devlin et al. (2019), we randomly apply the word-level masking to both inputs, $x^a$ and $x^b$. The goal is to force the model to preserve the contextual

information from the original sentences. Each word of the input sentence is randomly masked using the masking probability, $p_{mask}$. The second one is to encourage each hidden vector out of the encoder to be close to the origin, i.e., $\min \frac{\lambda}{M} \sum_{m=1}^{M} \sum_{i=1}^{L^m} \|h_i^m\|^2$, where $\lambda \geq 0$ is a regularization strength. Finally, we add small, zero-mean Gaussian noise to each vector $h_i^m$ before interpolation during training. The combination of these two has an effect of tightly packing all training inputs, or their hidden vectors induced by the encoder, in a small ball centered at the origin. In doing so, interpolation between two points passes by similar, interpolatable inputs placed in the hidden space. This is similar to how variational autoencoders form their latent variable space (Kingma & Welling, 2013).

### 3.4 Augmentation

Once trained, the model can be used to interpolate any pair of sentences with an arbitrary choice of decoding strategy and distribution of $\alpha$. When we are dealing with a single-sentence classification, we draw random pairs of training examples with their labels as a minibatch, $((x_m^a, y_m^a), (x_m^b, y_m^b))$ along with a randomly drawn mixup ratio value $\alpha_m$. Then, based on the learned distribution, LINDA generates an interpolated version $\tilde{x}$ where we set the corresponding interpolated label to $\tilde{y} = (1 - \alpha_m)y_m^a + \alpha_m y_m^b$. We further consider re-labeling under a self-training manner in §6.2. In the case where the input consists of two sentences, we produce an interpolated sentence for each of these sentences separately. For instance, in the case of natural language inference (NLI) (Rocktäschel et al., 2016), we interpolate the premise and hypothesis sentences separately and independently from each other and concatenate the interpolated premise and hypothesis to form an augmented training example.

## 4 LINDA: Training Details

We train LINDA once on a million sentence pairs randomly drawn from English Wikipedia and use it for all the experiments. We detail how we train this interpolation model in this section.

We base our interpolation model on a pre-trained BART released by Lewis et al. (2020). More specifically, we use BART-large. BART is a Transformer-based encoder-decoder model, pre-trained as a denoising autoencoder. We modify BART-large so that the encoder computes the hidden representations of two input sentences individually. These representations are up/downsampled to match the target interpolated length, after which they are linearly combined according to the mixing ratio. The decoder computes the interpolation distribution as in Eq. equation 2, in an autoregressive manner.

We train this model using the regularized reconstruction loss defined in §3.3. We uniformly sample the mixup ratio $\alpha \in [0, 1]$ at random for each minibatch during training as well as for evaluation. Once trained, we can produce an interpolated sentence from the model using decoding strategy. Details on training can be found in Appendix B.1.

## 5 Does LINDA Learn to Interpolate?

We first check whether LINDA preserves the contextual information of reference sentences proportionally to the mixup ratio $\alpha$, which is the key property of interpolation we have designed LINDA to exhibit. We investigate how much information from each of two original sentences is retained (measured in terms of SBERT (Reimers & Gurevych, 2019) cosine similarity as sentence similarity) while varying the mixup ratio $\alpha$. We expect a series of interpolated sentences to show a monotonically decreasing trend in SBERT similarity when computed against the first original sentence and the opposite trend against the second original sentence, as the mixup ratio moves from 1 to 0.

Figure 2 confirms our expectation and shows the monotonic trend. LINDA successfully reconstructs one of the original sentences with an extreme mixup ratio, with both unigram precision scores and SentenceBERT (SBERT) similarity close to $1$[2]. As $\alpha$ nears 0.5, the interpolated sentence deviates significantly from both of the input sentences, as anticipated. These interpolated sentences however continue to be well-formed, as

---

[2]We note that SBERT similarity between pair of original sentence is about 0.3

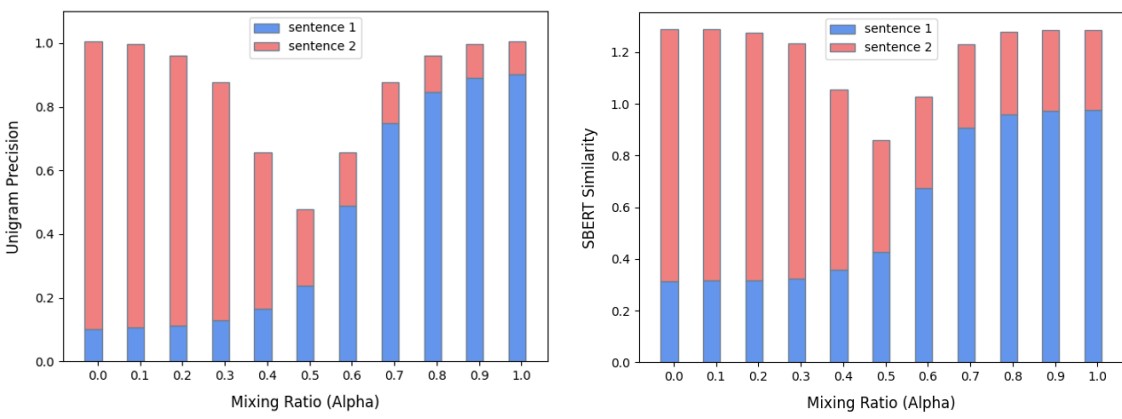

Figure 2: Average unigram precision (left) and semantic similarity (right) of the interpolated sentences while varying the mixup ratio $\alpha$ in Wikipedia

Table 1: Examples of interpolated sentences, generated by LINDA, and their original inputs on SST-2 and Yelp. Each background color of the text indicates different properties of the original sentences. ( blue : semantic property of input 1, green : semantic property of input 2, red : syntactic property)

| Mixup Ratio | | Examples |
|---|---|---|
| 0.50 | **Input 1** | Relaxed in its  perfect  quiet pace and proud  in its message |
| | **Input 2** | The plot is both  contrived and  cliched |
| | **LINDA** | The plot is both  perfect  and  cliched   in its message. |
| 0.60 | **Input 1** | Richly entertaining and  suggestive of any number of metaphorical readings |
| | **Input 2** | The film is  superficial and  will probably be of interest  primarily to  its target audience |
| | **LINDA** | The film is   very entertaining and   will probably be of interest   to  many  of   its target  audience  regardless of its genre. |
| 0.50 | **Input 1** | Loved the Vegan  menu! Glad to have so many options!  The free margarita  was really good  also! |
| | **Input 2** | Service was awfully slow  ...  food was just meh ....   eventhough we get comp for drinks  ... . the whole experienced didnt make up for it .. |
| | **LINDA** | Lincoln was very pleased with the  menu  ...   Glad to have so many options!  ...  eventhough we  got the free drinks  for the whole team ... it  was really good  .. |
| 0.54 | **Input 1** | engaging , imaginative filmmaking in its nearly 2 1/2 |
| | **Input 2** | the film  has ( its ) moments , but |
| | **LINDA** | The film  was released in the United States on October 2, 2019, and in Canada on October 3, 2019. |

we will show later for instance in Table 8 from Appendix E. This implies that LINDA indeed interpolates sentences over a natural language manifold composed mostly of well-formed sentences only, as we expected and designed it for. We observe a similar trend even when LINDA was used with non-Wikipedia sentences (see Appendix A for the other tasks).

In Table 1, we present four sample interpolations using four random pairs of validation examples from SST-2 (Socher et al., 2013) and Yelp[3] (see §6.1). As anticipated from the sentence similarity analysis above, most of

---

[3]https://www.yelp.com/dataset

the interpolated sentences exhibit some mix of semantic and syntactic structures from both of the original sentences, according to the mixing ratio, as well-demonstrated in the first and third examples. Especially in the third example, we observe both mixing of semantic and syntactic features as well as paraphrasing. The second example shows the LINDA's capability of paraphrasing as well as using synonyms. These examples provide evidence that LINDA has successfully learned to mix contents from two inputs, i.e., interpolate them, over a natural language manifold of well-formed sentences. Despite LINDA's ability to interpolate text, if both of the inputs are not well-formed, unlike those sentences in Wikipedia as in the final example, LINDA fails to interpolate these two ill-formed sentences and resorts to producing a Wikipedia-style, well-formed sentence, which clearly lies on a natural language manifold, that has only a vague overlap with either of the input sentences. However, we expect to mitigate existing risks by further training LINDA on additional datasets with diverse structures.

Overall, we find the observations in this section are evidence supporting that the proposed approach learns to interpolate between two sentences over a natural language manifold composed of well-formed sentences. In particular, LINDA is capable of generating syntactic samples that possess the sub-part or semantics of original phrases, but are distinct from the original sentences. Whereas existing interpolation approaches that apply the idea of mixup to NLP are almost never well-formed and cannot mix the original sentences in non-trivial ways.

## 6   LINDA for Data Augmentation

Encouraged by the promising interpolation results from the previous section, we conduct an extensive set of experiments to verify the effectiveness of LINDA for data augmentation in this section.

We run all experiments in this section by fine-tuning a pre-trained BERT-base (Devlin et al., 2019) classifier. We report the average accuracy and standard deviation over five runs. For each training example, we produce one perturbed example from a data augmentation method so that the size of the augmented dataset matches that of the original training set. Details on training can be found in Appendix B.2.

### 6.1   Downstream Tasks

We conduct experiments on eight datasets: TREC (coarse, fine) (Li & Roth, 2002), SST-2, IMDB (Maas et al., 2011), RTE (Wang et al., 2018), MNLI (Williams et al., 2018), QNLI (Rajpurkar et al., 2016) and MRPC (Dolan & Brockett, 2005). For four text classification datasets, we create two versions of training scenarios; a **full** setting where the original training set is used, and a **low**-resource setting where only 5 or 10 randomly selected training examples per class are used. We also test LINDA on OOD generalization, by closely following the protocol from Ng et al. (2020). We use the out-of-domain datasets of MNLI, Yelp Review and Amazon Review (AR) (Ni et al., 2019) datasets as well as the Movies dataset created by combining SST-2 and IMDB. We train a model separately on each domain and evaluate it on all the domains. We report accuracy using the official test set on TREC, IMDB, AR and YELP. For SST-2, RTE, MNLI, QNLI and MRPC, we use the validation set. Note that MNLI, QNLI, Yelp, AR training set are sampled randomly from a larger dataset. Refer to Appendix B.3 for more information.

### 6.2   Label Generation for LINDA

As our approach is under the unsupervised learning scheme, where only contextual information is considered, interpolated labels suggested in 3.4 have risk of being mismatched with the correct label. Therefore, we investigate using the predicted label which is evaluated with the model taught with the original training data, which we call *teacher*, to makes the model to be susceptible to the damaged label issue. Once the teacher model is trained on the original set of unaugmented data, we evaluate augmented samples and replace the label information $\tilde{y}$ with the output probability of a trained teacher model. We test three labeling strategies for LINDA and report the accuracy of two of them. For further details, refer to Appendix C.

Table 2: Comparison of classification accuracy (%) of $BERT_{BASE}$ after fine-tuning with each data augmentation method on four different datasets under full and low resource (5-shot/10-shot) settings. All the results are reported as "mean($\pm$std)" across 5 random runs. The best results are indicated in bold.

| Data Profile | Method | TREC-Coarse | TREC-Fine | SST-2 | IMDB |
|---|---|---|---|---|---|
| **Low** | Vanilla | $61.2_{\pm7.63}$ / $81.0_{\pm1.99}$ | $53.7_{\pm2.46}$ / $66.8_{\pm2.33}$ | $59.8_{\pm6.75}$ / $63.1_{\pm4.24}$ | $61.8_{\pm2.41}$ / $67.8_{\pm2.94}$ |
| | EDA | $57.0_{\pm6.54}$ / $72.0_{\pm4.64}$ | $\mathbf{54.7}_{\pm1.73}$ / $66.5_{\pm4.19}$ | $62.3_{\pm3.18}$ / $60.5_{\pm2.81}$ | $62.5_{\pm3.00}$ / $69.0_{\pm3.04}$ |
| | SSMBA | $60.0_{\pm5.38}$ / $80.6_{\pm1.53}$ | $53.9_{\pm1.42}$ / $66.5_{\pm4.66}$ | $60.3_{\pm6.26}$ / $64.8_{\pm5.78}$ | $64.1_{\pm2.10}$ / $69.9_{\pm2.63}$ |
| | Mixup | $59.9_{\pm4.94}$ / $81.6_{\pm2.57}$ | $52.3_{\pm2.12}$ / $67.9_{\pm3.03}$ | $59.7_{\pm4.64}$ / $62.1_{\pm4.38}$ | $62.2_{\pm1.86}$ / $67.4_{\pm2.37}$ |
| | LINDA | $\mathbf{62.2}_{\pm3.09}$ / $\mathbf{83.4}_{\pm2.30}$ | $54.2_{\pm1.50}$ / $\mathbf{69.2}_{\pm2.60}$ | $\mathbf{63.5}_{\pm4.44}$ / $\mathbf{66.5}_{\pm3.84}$ | $\mathbf{67.3}_{\pm2.60}$ / $\mathbf{71.0}_{\pm1.61}$ |
| **Full** | Vanilla | $97.2_{\pm0.36}$ | $87.9_{\pm1.05}$ | $92.3_{\pm0.06}$ | $89.2_{\pm0.59}$ |
| | EDA | $96.4_{\pm0.77}$ | $90.6_{\pm0.84}$ | $92.2_{\pm0.52}$ | $\mathbf{89.5}_{\pm0.18}$ |
| | SSMBA | $96.9_{\pm0.41}$ | $90.2_{\pm0.71}$ | $92.3_{\pm0.38}$ | $89.2_{\pm0.17}$ |
| | $SSMBA_{soft}$ | $97.3_{\pm0.23}$ | $88.3_{\pm0.54}$ | $92.3_{\pm0.35}$ | $89.4_{\pm0.13}$ |
| | Mixup | $97.5_{\pm0.30}$ | $86.4_{\pm1.23}$ | $92.4_{\pm0.41}$ | $89.1_{\pm0.18}$ |
| | LINDA | $\mathbf{97.6}_{\pm0.36}$ | $\mathbf{91.4}_{\pm0.26}$ | $92.3_{\pm0.43}$ | $89.2_{\pm0.15}$ |
| | $LINDA_{soft}$ | $97.4_{\pm0.09}$ | $90.1_{\pm0.39}$ | $\mathbf{92.8}_{\pm0.34}$ | $89.3_{\pm0.18}$ |

Table 3: Performance on three NLI datasets (RTE, QNLI, MNLI) and one paraphrase identification dataset (MRPC).

| Method | RTE | QNLI | MNLI (m) | MRPC |
|---|---|---|---|---|
| Vanilla | $68.2_{\pm1.78}$ | $88.0_{\pm0.77}$ | $77.0_{\pm0.50}$ | $84.0_{\pm0.57}$ |
| EDA | $65.6_{\pm2.22}$ | $86.6_{\pm0.70}$ | $76.4_{\pm0.27}$ | $81.9_{\pm1.17}$ |
| SSMBA | $66.8_{\pm2.18}$ | $87.6_{\pm0.58}$ | $76.4_{\pm1.18}$ | $83.7_{\pm0.82}$ |
| $SSMBA_{soft}$ | $69.4_{\pm2.36}$ | $88.5_{\pm1.08}$ | $77.0_{\pm0.10}$ | $83.8_{\pm0.27}$ |
| Mixup | $70.9_{\pm0.94}$ | $87.6_{\pm1.13}$ | $76.8_{\pm0.22}$ | $84.2_{\pm0.42}$ |
| LINDA | $66.1_{\pm3.83}$ | $87.6_{\pm0.45}$ | $76.3_{\pm0.27}$ | $82.5_{\pm0.36}$ |
| $LINDA_{soft}$ | $\mathbf{71.9}_{\pm1.87}$ | $\mathbf{88.6}_{\pm0.25}$ | $\mathbf{77.4}_{\pm0.27}$ | $\mathbf{85.2}_{\pm0.17}$ |

### 6.3 Baselines

We compare LINDA against three existing augmentation methods. First, EDA[4](Wei & Zou, 2019a) is a heuristic method that randomly performs synonym replacement, random insertion, random swap, and random deletion. Second, SSMBA[5] (Ng et al., 2020) perturbs an input sentence by letting a masked language model reconstruct a randomly corrupted input sentence. For SSMBA, we report two different performances; one (**SSMBA**) where the label of an original input is used as it is, and the other (**SSMBA**$_{soft}$) where a predicted categorical distribution (soft label) by another classifier trained on an original, unaugmented dataset is used. Third, mixup (Guo et al., 2019) mixes a pair of sentence at the sentence embedding level. Note that we report accuracies of our proposed method in both LINDA and LINDA$_{soft}$ settings, where LINDA$_{soft}$ is a counterpart of SSMBA$_{soft}$.

### 6.4 Result

In Table 2 we present the results on four text classification datasets for both full and low-resource settings. In the case of **low**-resource settings, LINDA outperforms all the other augmentation methods except for the 5-shot setting with the TREC-fine task, although the accuracies are all within standard deviations in this particular case. Because Ng et al. (2020) reported low accuracies when predicted labels were used with a

---

[4]https://github.com/jasonwei20/eda_nlp
[5]https://github.com/nng555/ssmba

Table 4: Performance on out-of-domain tasks.

| Method | Movie | Yelp | $AR_{clothing}$ | MNLI (mm) |
|---|---|---|---|---|
| Vanilla | $84.8_{\pm0.74}$ | $65.1_{\pm1.47}$ | $72.0_{\pm2.79}$ | $77.6_{\pm0.46}$ |
| EDA | $84.8_{\pm0.88}$ | $64.5_{\pm1.32}$ | $71.6_{\pm2.84}$ | $77.1_{\pm0.39}$ |
| SSMBA | $84.7_{\pm0.79}$ | $64.9_{\pm1.26}$ | $71.6_{\pm2.85}$ | $77.3_{\pm0.51}$ |
| $SSMBA_{soft}$ | $85.5_{\pm0.86}$ | $\mathbf{65.4}_{\pm1.36}$ | $\mathbf{73.3}_{\pm2.87}$ | $77.7_{\pm0.29}$ |
| Mixup | $83.9_{\pm1.71}$ | $65.0_{\pm1.20}$ | $72.0_{\pm2.83}$ | $77.6_{\pm0.31}$ |
| LINDA | $85.1_{\pm0.97}$ | $64.7_{\pm1.31}$ | $71.7_{\pm3.11}$ | $77.1_{\pm0.19}$ |
| $LINDA_{soft}$ | $\mathbf{85.7}_{\pm0.86}$ | $\mathbf{65.4}_{\pm1.08}$ | $72.0_{\pm2.89}$ | $\mathbf{78.0}_{\pm0.29}$ |

small number of training examples already, we do not test $SSMBA_{soft}$ and $LINDA_{soft}$ in the low-resource settings. Similarly in the full setting, our proposed model outperforms all the other methods on all datasets except for IMDB, although the accuracies are all within two standard deviations. More specifically, LINDA shows the best results for TREC-Coarse and TREC-Fine whereas $LINDA_{soft}$ shows the best performance in SST-2. This demonstrates the effectiveness of the proposed approach regardless of the availability of labeled examples.

In the first three columns of Table 3, we report accuracies on the NLI and paraphrase identification datasets; RTE, QNLI, MNLI (matched) and MRPC and show that $LINDA_{soft}$ outperforms other data augmentation methods on all of the four datasets. $LINDA_{soft}$ improves the evaluation accuracy by 5.4% on RTE, 0.7% on QNLI, 0.5% on MNLI (matched) and 1.2% on MPRC when all the other augmentation methods, except for $SSMBA_{soft}$, degrade the Vanilla performance.

Table 4 shows the OOD generalization accuracies on Movie, Yelp, $AR_{clothing}$ and MNLI (mismatched). Although $SSMBA_{soft}$ achieves the greatest improvement in OOD generalization over the vanilla approach in Yelp and $AR_{clothing}$, $LINDA_{soft}$ does not fail to show improvement over the baseline across all datasets in both in-domain and OOD settings. In agreement with Ng et al. (2020), we could not observe any improvement with the other augmentation methods, EDA and Mixup in this experiment.

## 7 Conclusion

Mixup, which was originally proposed for computer vision applications, has been adapted in recent years for the problems in NLP. To address the issues of arising from applying mixup to variable-length discrete sequences, existing efforts have largely relied on heuristics-based interpolation. In this paper, we take a step back and started by contemplating what it means to interpolate two sequences of different lengths and discrete tokens. We then come up with a set of conditions that should be satisfied by an interpolation operator on natural language sentences and proposed a neural net based interpolation scheme, called LINDA, that stochastically produces an interpolated sentence given a pair of input sentences and a mixing ratio.

We empirically show that LINDA is indeed able to interpolate two natural language sentences over a natural language manifold of well-formed sentences both quantitatively and qualitatively. We then test it for the original purpose of data augmentation by plugging it into mixup and testing it on nine different datasets. LINDA outperforms all the other data augmentation methods, including EDA and SSMBA, on most datasets and settings, and consistently outperforms non-augmentation baselines, which is not the case with the other augmentation methods.

## 8 Limitations

Despite the recent success of deep learning for natural language processing and language generation, there is a major limitation to any language generation system based on deep learning, that is, the lack of controllability. Because LINDA is built on top of latest advances in deep learning as well as its applications to natural

language processing and generation, LINDA also lacks controllability. More specifically for LINDA, this lack of controllability implies that we, as users of LINDA, cannot explicitly select nor exclude different ways by which a given pair of text are interpolated. Some ways to interpolate may be more desirable than the others, but we cannot really choose which is chosen by LINDA. More generally, this lack of controllability prevents us from avoiding or preventing undesirable biases (Sun et al., 2019) in a dataset, or statistical regularities from a large, underlying corpus from being captured and used for interpolation. We believe the community should focus increasingly more on addressing this lack of controllability in neural language generation, which will be directly applicable to the proposed approach in this paper.

## Broader Impact and Ethical Implications

The improvement by LINDA on low-resource scenarios as well as OOD generalization suggests that LINDA can facilitate more broader adoption of advanced natural language processing (NLP) techniques beyond traditionally well-served/studied languages. Despite such improvement, LINDA does not resolve potentially negative societal impacts caused by underlying NLP systems, such as gender bias (Bordia & Bowman, 2019) in a dataset. Since data augmentation is typically based on pre-defined fixed augmentations, it is challenging to explicitly remove such inherency of undesirable properties. However, it can be mitigated by giving a higher mixing ratio to unbiased sentences when generating an interpolated sentence. Hence, we believe that our method could contribute to augmenting fair data for NLP task.

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

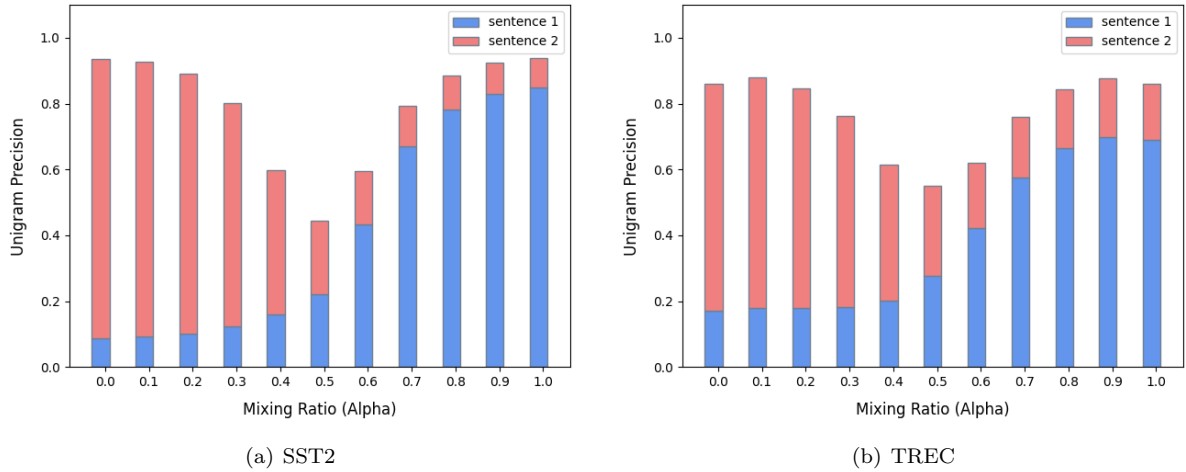

(a) SST2        (b) TREC

Figure 3: Average unigram precision score of the interpolated sentences with varying mixup ratio values

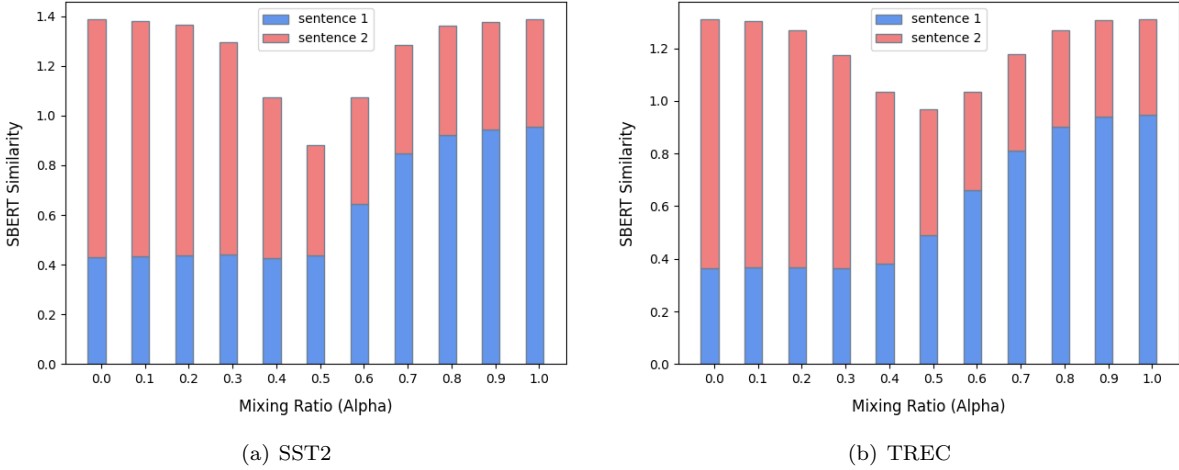

(a) SST2        (b) TREC

Figure 4: Average semantic similarity of the interpolated sentences with varying mixup ratio values

## A   Interpolation on Different Domain

Figure 4 shows the average sentence similarity of the interpolated sentences with respect to the mixup ratio, $\alpha$. We observe that the general trend is identical to the average sentence similarity measured in the Wikipedia corpus, as presented in Figure 2. Sentence similarity between the original pair of the sentence is 0.44 for SST2 and 0.38 for the TREC dataset. We also evaluated with average unigram precision score of the interpolated sentences in the same way in Figure 3. As can be observed, similar trend even when LINDA was used with non-Wikipedia sentences, although overall average unigram precision scores across the mixing ratios are generally a little bit lower than they were with Wikipedia.

# B Details of Experiment

## B.1 Hyperparameters for LINDA

Our model based on BART-large model have 400M parameters. We set the batch size to 8 and use Adam optimizer (Kingma & Ba, 2015) with a fixed learning rate of $1e^{-5}$ same as (Kumar et al., 2020a). We use 8 GPUs (Tesla T4) for training. A word-level masking is adapt to input sentence with $p_{\text{mask}} = 0.1$. We set $\lambda = 0.001$ and the standard deviation of Gaussian noise to 0.001. For decoding strategy, we use beam search with beam size set to 4 in all the experiments.

## B.2 Hyperparameters for Downstream Tasks

We experiment with BERT-base model which has 110M parameters provided by HuggingFace Transformers (Wolf et al., 2019) for downstream tasks. We use Adam (Kingma & Ba, 2015) with a fixed learning rate of $3e^{-5}$. We use the batch size of 32 and train a model for 5 epochs when we use a full task dataset, and the batch size of 8 for 30 epochs in the case of low-resource settings.

## B.3 Datasets for Downstream Tasks

| Dataset | Task | Label | Train |
|---|---|---|---|
| $\text{TREC}_{coarse}$ | Classification | 6 | 5.5k |
| $\text{TREC}_{fine}$ | Classification | 47 | 5.5k |
| SST-2 | Sentiment Analysis | 2 | 67k |
| IMDB | Sentiment Analysis | 2 | 46k |
| RTE | NLI | 2 | 2.5k |
| MNLI | NLI | 3 | 25k$^\dagger$ |
| QNLI | NLI | 2 | 25k$^\dagger$ |
| MRPC | Paraphrase | 2 | 4.0k |
| $\text{AR}_{\text{clothing}}$ | Rating Prediction | 5 | 25k$^\dagger$ |
| Yelp | Rating Prediction | 5 | 25k$^\dagger$ |

Table 5: Dataset summary statistics. *C*, *S*, *OOD*, and, *NLI* in the task column represents classification, sentiment analysis, out-of-distribution, and natural language inference respectively. Training sets marked with a † are sampled randomly from a larger dataset.

We present the details of downstream datasets in Table 5. For experiments, we report accuracy using the official test set on TREC, IMDB, AR and YELP. For those where the label for the official test set is not available, SST-2, RTE, MNLI, QNLI and MRPC, we use the validation set as the test set. Further details are shown in Table 6. SST-2 and RTE are from the GLUE dataset (Wang et al., 2018). We report the number of the samples in each split in the (train/validation) format. Rows for TREC and IMDB are reported in the (train/test) format. MNLI, QNLI, $\text{AR}_{\text{clothing}}$ and Yelp can be downloaded at https://nyu.box.com/s/henvmy17tkyr6npl7e1ltw8j46baxsml. Other datasets can be downloaded at HuggingFace[6]. For OOD experiments, we use four different datasets; MNLI, $\text{AR}_{\text{clothing}}$, Yelp, Movies. $\text{AR}_{\text{clothing}}$ consists with five clothing categories: *Clothes, Women clothing, Men Clothing, Baby Clothing, Shoes* and Yelp, following the preprocessing in Hendrycks et al. (2020), with four groups of food types: *American, Chinese, Italian,* and *Japanese.* For both datasets, we sample 25,000 reviews for each category and the model predicts a review's 1 to 5 star rating. Movie dataset is consists with SST-2 and IMDB. We train the model separately on each domain, then evaluate on all domains to report the average in-domain and OOD performance across 5 runs.

| Dataset | | # |
|---|---|---|
| **TREC**$_{coarse}$ | | 5.5k / 500 |
| **TREC**$_{fine}$ | | 5.5k / 500 |
| **SST-2** | | 67k / 872 |
| **IMDB** | | 25k / 25k |
| **RTE** | | 2.5k / 3k |
| **MNLI** | | 2.5k / 9.7k / 9.8k |
| **QNLI** | | 2.5k / 5.5k |
| **MRPC** | | 4.0k / 1.7k |
| **AR**$_{\text{clothing}}$ | Clothing | 25k / 2k |
| | Women | 25k / 2k |
| | Men | 25k/ 2k |
| | Baby | 25k/ 2k |
| | Shoes | 25k / 2k |
| **YELP** | American | 25k / 2k |
| | Chinese | 25k / 2k |
| | Italian | 25k / 2k |
| | Japanese | 25k / 2k |

Table 6: Summary of which split of dataset is used to report the performance on downstream datasets. Column # indicates the number of samples in the data split as either in (train/validation) or in (train/test) based on the split used to report the performance with. MNLI has two test data for domain matched (in-domain) and domain mismatched (out-of-domain) settings, respectively.

| Label | $T$ | TREC-Coarse | TREC-Fine | SST-2 | IMDB | RTE | QNLI | MNLI-(m/mm) |
|---|---|---|---|---|---|---|---|---|
| Interpolated | 1 | **97.6**$_{\pm 0.36}$ | 91.4$_{\pm 0.26}$ | 92.3$_{\pm 0.43}$ | 89.2$_{\pm 0.15}$ | 66.1$_{\pm 3.8}$ | 88.0$_{\pm 0.4}$ | 77.0$_{\pm 0.3}$/77.1$_{\pm 0.2}$ |
| Interpolated | 0.5 | 97.4$_{\pm 0.21}$ | **91.6**$_{\pm 0.75}$ | 92.2$_{\pm 0.37}$ | 89.3$_{\pm 0.22}$ | 66.8$_{\pm 1.4}$ | 87.4$_{\pm 0.7}$ | 76.2$_{\pm 0.4}$/77.1$_{\pm 0.4}$ |
| predicted | - | 97.4$_{\pm 0.09}$ | 90.1$_{\pm 0.39}$ | **92.8**$_{\pm 0.34}$ | **89.3**$_{\pm 0.18}$ | **71.9**$_{\pm 1.9}$ | **88.6**$_{\pm 0.2}$ | **77.4**$_{\pm 0.3}$/**78.0**$_{\pm 0.3}$ |

Table 7: Effect of using the different label generation

## C  Label Generation for LINDA

### C.1  Choice of the Label Generation

For the full dataset setting, shown in Table 2, evaluation accuracy of LINDA is reported with different labeling method for each dataset. Note that we only report accuracy with the interpolated labels in the low-resources settings since Ng et al. (2020) has already reported that with a small amount of training examples, using the psuedo labeling show low accuracies. Using the interpolated labels for TREC-coarse and TREC-Fine shows the best performances in the full dataset setting. However, for IMDB and SST-2, utilizing the predicted soft label shows the best performance for the full setting experiment.

For NLI experiments, reported in Table 3, best performances for all three datasets are also with the predicted soft label. Lastly, for the OOD experiments in Table 4, using the predicted soft label shows the best performance in MNLI, AR$_{\text{clothing}}$ and Yelp datasets across all domains (five different domains in AR$_{\text{clothing}}$ and four different domains in Yelp). In Movie, using the interpolated labels explained in §3.4 shows the best result when training the model on IMDB and evaluating on SST-2. However, when training the model on SST-2 and evaluating on IMDB, using the predicted soft label shows the best result.

### C.2  Effect of Label Sharpening

Although we apply linear interpolation in LINDA, we observed that unigram precision score of interpolated sentence is on the side of the larger $\alpha$ value well Figure 2 in §5. Based on this observation, we investigate the

---

[6]https://huggingface.co/datasets

effect of sharpening our label $\tilde{y}$ by applying the *sharpening* with temperature $T$ (Heaton, 2017). Denote $\tilde{y}$ is soft label of a class distributions $\tilde{y}_i$ and sharpening operation is defined as operation.

$$Sharpen(\tilde{y_m}, T) = \tilde{y}_i^{1/T} / \sum_{i=1}^{L} exp(\tilde{y}_i^{1/T}), \tag{6}$$

When $T$ equals 1, the label does not change and as $T$ gets closer to 0, $Sharpen(\tilde{y}, T)$ approaches to a hard label. We show the effect of using sharpened labels in Table 7.

### C.3 Effect of the Predicted Label

In Table 7, we show the difference in accuracy when the interpolated label is used versus when the predicted label is used. It shows that using the predicted label significantly improves the performance in SST-2 and NLI datasets, despite of using the interpolated label even hurting the performance.

## D ID and OOD Results over All Domains

Table 4 in §6.4 shows the average OOD accuracy (%) over all categories. However, unlike the Movie dataset, where it consists of only two domains; SST-2 and IMDB, $AR_{clothing}$ and Yelp datasets consist of multiple domains. $AR_{clothing}$ consists with five clothing categories: *Clothes, Women clothing, Men Clothing, Baby Clothing, Shoes* and Yelp, following the preprocessing in Hendrycks et al. (2020), with four groups of food types: *American, Chinese, Italian,* and *Japanese.* In MNLI, We train on the 5 genres with training data and test with data have different genres. We report accuracy (%) over all categories along with standard deviation values over 5 random runs in the $AR_{clothing}$ and Yelp datasets in Table 9 and Table 10.

## E Effect of the Mixup Ratio with Wikipedia Sentences

We present an example of the interpolated sentences from Wikipedia sentences with the varying values of *alpha* in Table 8. Two reference sentences are randomly drawn from English Wikipedia corpus. As *alpha* gets closer to 1, the interpolated sentence resembles the sentence 1 more and as *alpha* gets closer to 0, it resembles the sentence 2 more. Moreover, we see that LINDA is capable of generating sentences using tokens that have never appeared in the reference sentences as well. This capability of LINDA comes from leveraging the power of pre-trained generation model, BART, on the vast amount of English Wikipedia sentences with the reconstruction loss.

| Sentence 1 | Sentence 2 |
|---|---|
| Henry tried, with little success, to reacquire the property Richard had sold, and had to live modestly as a gentle man, never formally taking title to the earldom | Kobilje Creek, a left tributary of the Ledava River, flows through it. |

| $\alpha$ | Generated Sentences |
|---|---|
| 0.1 | Kobilje Creek, a left tributary of the Ledava River, flows through it. |
| 0.2 | Kobilje Creek, a left tributary of the Ledava River, flows through it to the north. |
| 0.3 | Kobilje Creek, a left tributary of the River Ledava, flows through it, as does a small river. |
| 0.4 | Kobilje, a small creek, left the tributary of the Ledava River, and it flows through the town. |
| 0.5 | Kenny, with a little help from his father, was able to take over the family business, which had been run by his mother. |
| 0.6 | Henry tried, with little success, to reacquaint himself with the property, and had to live as a modest earl. |
| 0.7 | Henry tried, with little success, to reacquaint himself wit h the property, and had to live as a modest gentleman, never formally taking part in the title. |
| 0.8 | Henry tried, with little success, to reacquaint himself with the property, and had to live as a modest gentleman, never formally taking title to the earldom |
| 0.9 | Henry tried, with little success, to reacquire the property Richard had sold, and had to live modestly as a gentleman, never formally taking title to the title. |

Table 8: Example of a series of interpolated sentences by adjusting the value of *alpha* given two input sentences.

| *Vanilla* | Clothing | Women | Men | Baby | Shoes |
|---|---|---|---|---|---|
| **Clothing** | $72.1_{\pm0.40}$ | $71.8_{\pm0.29}$ | $72.0_{\pm0.49}$ | $71.3_{\pm0.49}$ | $71.6_{\pm0.73}$ |
| **Women** | $69.6_{\pm0.62}$ | $69.5_{\pm0.29}$ | $68.2_{\pm0.50}$ | $67.7_{\pm0.96}$ | $68.2_{\pm0.79}$ |
| **Men** | $71.9_{\pm0.50}$ | $72.0_{\pm1.15}$ | $72.6_{\pm0.50}$ | $71.4_{\pm0.98}$ | $71.2_{\pm0.11}$ |
| **Baby** | $76.9_{\pm0.39}$ | $76.4_{\pm0.28}$ | $76.9_{\pm0.36}$ | $78.2_{\pm0.24}$ | $0.77_{\pm0.93}$ |
| **Shoes** | $71.8_{\pm0.67}$ | $71.4_{\pm0.72}$ | $72.3_{\pm0.37}$ | $70.5_{\pm0.16}$ | $73.0_{\pm0.43}$ |

| *EDA* | Clothing | Women | Men | Baby | Shoes |
|---|---|---|---|---|---|
| **Clothing** | $71.9_{\pm0.71}$ | $71.0_{\pm0.69}$ | $71.8_{\pm0.56}$ | $70.9_{\pm0.45}$ | $71.8_{\pm0.56}$ |
| **Women** | $68.3_{\pm1.35}$ | $69.3_{\pm0.38}$ | $67.9_{\pm0.60}$ | $67.3_{\pm1.06}$ | $67.5_{\pm0.44}$ |
| **Men** | $71.6_{\pm0.62}$ | $71.2_{\pm0.76}$ | $71.8_{\pm0.34}$ | $71.1_{\pm0.66}$ | $72.1_{\pm0.37}$ |
| **Baby** | $76.6_{\pm0.63}$ | $76.4_{\pm0.35}$ | $76.0_{\pm0.37}$ | $77.6_{\pm0.38}$ | $76.4_{\pm0.57}$ |
| **Shoes** | $71.2_{\pm0.68}$ | $71.5_{\pm0.48}$ | $71.3_{\pm0.52}$ | $70.8_{\pm0.52}$ | $72.7_{\pm0.26}$ |

| *SSMBA* | Clothing | Women | Men | Baby | Shoes |
|---|---|---|---|---|---|
| **Clothing** | $71.9_{\pm0.60}$ | $71.6_{\pm1.18}$ | $71.9_{\pm0.71}$ | $70.3_{\pm0.67}$ | $71.5_{\pm0.84}$ |
| **Women** | $68.7_{\pm0.60}$ | $69.2_{\pm0.94}$ | $67.9_{\pm0.84}$ | $67.4_{\pm0.64}$ | $67.9_{\pm1.16}$ |
| **Men** | $72.0_{\pm0.60}$ | $71.5_{\pm1.46}$ | $72.1_{\pm0.63}$ | $70.9_{\pm1.06}$ | $71.3_{\pm1.01}$ |
| **Baby** | $76.7_{\pm0.68}$ | $76.3_{\pm0.77}$ | $75.9_{\pm0.64}$ | $77.7_{\pm0.26}$ | $76.4_{\pm1.02}$ |
| **Shoes** | $71.7_{\pm0.49}$ | $72.0_{\pm0.41}$ | $71.1_{\pm0.87}$ | $70.3_{\pm0.83}$ | $72.7_{\pm0.48}$ |

| *SSMBA*$_{\text{soft}}$ | Clothing | Women | Men | Baby | Shoes |
|---|---|---|---|---|---|
| **Clothing** | $73.2_{\pm0.49}$ | $72.4_{\pm0.48}$ | $72.0_{\pm0.32}$ | $70.7_{\pm0.40}$ | $72.0_{\pm0.64}$ |
| **Women** | $69.0_{\pm0.44}$ | $70.0_{\pm0.34}$ | $68.9_{\pm0.79}$ | $68.4_{\pm0.45}$ | $68.1_{\pm0.77}$ |
| **Men** | $72.2_{\pm0.56}$ | $72.3_{\pm0.49}$ | $72.8_{\pm0.42}$ | $70.9_{\pm0.55}$ | $71.5_{\pm1.01}$ |
| **Baby** | $77.6_{\pm0.42}$ | $76.8_{\pm0.58}$ | $77.3_{\pm0.63}$ | $78.6_{\pm0.44}$ | $76.7_{\pm0.87}$ |
| **Shoes** | $71.9_{\pm0.47}$ | $72.4_{\pm0.52}$ | $72.3_{\pm0.89}$ | $70.8_{\pm0.64}$ | $72.9_{\pm0.37}$ |

| *Mixup* | Clothing | Women | Men | Baby | Shoes |
|---|---|---|---|---|---|
| **Clothing** | $72.2_{\pm0.53}$ | $72.0_{\pm1.45}$ | $71.4_{\pm0.43}$ | $71.2_{\pm0.56}$ | $71.8_{\pm0.34}$ |
| **Women** | $69.1_{\pm0.49}$ | $69.8_{\pm0.57}$ | $68.0_{\pm1.10}$ | $68.0_{\pm0.68}$ | $68.5_{\pm0.41}$ |
| **Men** | $71.5_{\pm0.44}$ | $72.7_{\pm0.72}$ | $72.5_{\pm0.14}$ | $71.0_{\pm0.63}$ | $72.3_{\pm0.47}$ |
| **Baby** | $76.7_{\pm1.18}$ | $77.2_{\pm0.24}$ | $76.4_{\pm0.23}$ | $78.0_{\pm0.21}$ | $76.8_{\pm0.45}$ |
| **Shoes** | $71.9_{\pm0.56}$ | $72.2_{\pm0.80}$ | $72.0_{\pm0.79}$ | $70.0_{\pm0.50}$ | $73.1_{\pm0.24}$ |

| *LINDA* | Clothing | Women | Men | Baby | Shoes |
|---|---|---|---|---|---|
| **Clothing** | $71.5_{\pm0.44}$ | $71.4_{\pm1.19}$ | $70.9_{\pm0.56}$ | $70.0_{\pm1.12}$ | $70.8_{\pm1.48}$ |
| **Women** | $68.6_{\pm0.24}$ | $70.3_{\pm1.23}$ | $67.9_{\pm0.50}$ | $66.8_{\pm1.32}$ | $67.6_{\pm1.31}$ |
| **Men** | $71.8_{\pm0.29}$ | $72.4_{\pm0.45}$ | $72.3_{\pm0.41}$ | $70.8_{\pm0.60}$ | $70.9_{\pm1.16}$ |
| **Baby** | $77.4_{\pm0.33}$ | $76.8_{\pm0.28}$ | $76.5_{\pm0.28}$ | $78.0_{\pm0.22}$ | $76.8_{\pm0.85}$ |
| **Shoes** | $71.8_{\pm0.54}$ | $72.1_{\pm0.68}$ | $71.5_{\pm1.14}$ | $70.9_{\pm0.98}$ | $73.2_{\pm0.48}$ |

| *LINDA*$_{\text{soft}}$ | Clothing | Women | Men | Baby | Shoes |
|---|---|---|---|---|---|
| **Clothing** | $72.9_{\pm0.32}$ | $72.1_{\pm0.60}$ | $71.8_{\pm0.38}$ | $70.9_{\pm0.42}$ | $71.6_{\pm0.39}$ |
| **Women** | $69.4_{\pm0.59}$ | $70.0_{\pm0.41}$ | $68.4_{\pm0.46}$ | $68.0_{\pm1.17}$ | $68.1_{\pm0.39}$ |
| **Men** | $71.7_{\pm0.31}$ | $72.4_{\pm0.45}$ | $72.8_{\pm0.37}$ | $70.3_{\pm1.15}$ | $71.6_{\pm0.80}$ |
| **Baby** | $77.1_{\pm0.29}$ | $76.8_{\pm0.36}$ | $77.2_{\pm0.41}$ | $78.3_{\pm0.42}$ | $76.9_{\pm0.39}$ |
| **Shoes** | $72.5_{\pm0.46}$ | $71.6_{\pm0.74}$ | $72.0_{\pm0.26}$ | $70.5_{\pm0.84}$ | $73.3_{\pm0.52}$ |

Table 9: Accuracy (%) of BERT$_{\text{BASE}}$ in the $\mathbf{AR}_{\text{clothing}}$ dataset. Each column represents the category where a model is trained with and each row represents the category where a model is evaluated on. All the results are reported as "mean(±std)" across 5 random runs.

| *Vanilla* | American | Chinese | Italian | Japanese |
|---|---|---|---|---|
| **American** | $66.2_{\pm 0.49}$ | $65.2_{\pm 0.67}$ | $64.5_{\pm 0.96}$ | $65.6_{\pm 1.51}$ |
| **Chinese** | $65.3_{\pm 0.71}$ | $64.6_{\pm 0.39}$ | $64.4_{\pm 0.64}$ | $65.7_{\pm 0.65}$ |
| **Italian** | $66.8_{\pm 0.52}$ | $65.7_{\pm 0.40}$ | $66.4_{\pm 0.34}$ | $65.8_{\pm 0.55}$ |
| **Japanese** | $65.5_{\pm 1.34}$ | $66.4_{\pm 0.99}$ | $64.9_{\pm 0.48}$ | $66.6_{\pm 0.52}$ |

| *EDA* | American | Chinese | Italian | Japanese |
|---|---|---|---|---|
| **American** | $65.5_{\pm 0.37}$ | $64.5_{\pm 1.49}$ | $64.3_{\pm 1.15}$ | $65.5_{\pm 0.32}$ |
| **Chinese** | $63.5_{\pm 0.51}$ | $63.7_{\pm 0.45}$ | $62.1_{\pm 1.08}$ | $63.6_{\pm 0.49}$ |
| **Italian** | $65.5_{\pm 0.66}$ | $64.4_{\pm 1.38}$ | $65.3_{\pm 0.74}$ | $64.3_{\pm 0.56}$ |
| **Japanese** | $65.9_{\pm 0.54}$ | $65.8_{\pm 0.76}$ | $64.3_{\pm 0.60}$ | $65.6_{\pm 0.45}$ |

| *SSMBA* | American | Chinese | Italian | Japanese |
|---|---|---|---|---|
| **American** | $65.2_{\pm 0.91}$ | $65.4_{\pm 0.99}$ | $64.8_{\pm 0.82}$ | $65.7_{\pm 0.36}$ |
| **Chinese** | $63.4_{\pm 0.70}$ | $63.8_{\pm 0.58}$ | $63.2_{\pm 0.81}$ | $63.6_{\pm 0.80}$ |
| **Italian** | $66.0_{\pm 0.76}$ | $65.3_{\pm 0.85}$ | $66.7_{\pm 0.75}$ | $65.3_{\pm 0.88}$ |
| **Japanese** | $64.6_{\pm 1.01}$ | $66.2_{\pm 1.39}$ | $65.3_{\pm 0.79}$ | $66.7_{\pm 0.33}$ |

| *SSMBA*$_{\text{soft}}$ | American | Chinese | Italian | Japanese |
|---|---|---|---|---|
| **American** | $66.6_{\pm 0.69}$ | $65.1_{\pm 0.88}$ | $65.1_{\pm 0.56}$ | $66.0_{\pm 0.92}$ |
| **Chinese** | $63.9_{\pm 1.02}$ | $65.1_{\pm 0.53}$ | $63.2_{\pm 0.61}$ | $64.3_{\pm 0.48}$ |
| **Italian** | $66.5_{\pm 0.46}$ | $65.6_{\pm 0.58}$ | $67.0_{\pm 0.29}$ | $66.4_{\pm 0.60}$ |
| **Japanese** | $66.1_{\pm 0.69}$ | $67.5_{\pm 0.98}$ | $64.8_{\pm 0.97}$ | $67.0_{\pm 0.34}$ |

| *Mixup* | American | Chinese | Italian | Japanese |
|---|---|---|---|---|
| **American** | $65.6_{\pm 0.59}$ | $65.0_{\pm 0.66}$ | $64.8_{\pm 1.10}$ | $65.6_{\pm 1.14}$ |
| **Chinese** | $63.5_{\pm 1.05}$ | $64.5_{\pm 0.58}$ | $63.1_{\pm 0.54}$ | $64.0_{\pm 0.45}$ |
| **Italian** | $66.0_{\pm 1.17}$ | $65.1_{\pm 0.63}$ | $66.3_{\pm 0.52}$ | $65.6_{\pm 0.51}$ |
| **Japanese** | $65.9_{\pm 0.72}$ | $66.0_{\pm 0.67}$ | $65.3_{\pm 0.93}$ | $66.2_{\pm 0.38}$ |

| *LINDA* | American | Chinese | Italian | Japanese |
|---|---|---|---|---|
| **American** | $65.3_{\pm 0.79}$ | $64.6_{\pm 1.53}$ | $64.4_{\pm 1.38}$ | $65.7_{\pm 0.52}$ |
| **Chinese** | $63.6_{\pm 0.71}$ | $64.0_{\pm 0.39}$ | $62.6_{\pm 0.64}$ | $63.6_{\pm 0.65}$ |
| **Italian** | $66.1_{\pm 0.79}$ | $65.0_{\pm 0.91}$ | $66.1_{\pm 0.21}$ | $65.2_{\pm 0.48}$ |
| **Japanese** | $65.8_{\pm 0.61}$ | $65.4_{\pm 0.97}$ | $64.1_{\pm 1.07}$ | $66.9_{\pm 0.36}$ |

| *LINDA*$_{\text{soft}}$ | American | Chinese | Italian | Japanese |
|---|---|---|---|---|
| **American** | $66.2_{\pm 0.72}$ | $65.7_{\pm 0.40}$ | $65.1_{\pm 0.55}$ | $66.2_{\pm 0.28}$ |
| **Chinese** | $64.2_{\pm 0.42}$ | $64.7_{\pm 0.51}$ | $63.8_{\pm 0.48}$ | $64.5_{\pm 1.01}$ |
| **Italian** | $66.2_{\pm 0.86}$ | $65.9_{\pm 0.44}$ | $67.0_{\pm 0.77}$ | $65.7_{\pm 0.86}$ |
| **Japanese** | $65.7_{\pm 0.56}$ | $67.0_{\pm 0.75}$ | $64.9_{\pm 0.70}$ | $66.8_{\pm 0.48}$ |

Table 10: Accuracy (%) of BERT$_{\text{BASE}}$ in the **Yelp** dataset. Each column represents the category where a model is trained with and each row represents the category where a model is evaluated on. All the results are reported as "mean($\pm$std)" across 5 random runs.

