# OpenReview forum: "LINDA: Unsupervised Learning to Interpolate in Natural Language Processing"
_TMLR — Rejected by TMLR_

### Review · Reviewer_jTWk · 2022-11-27

**Summary Of Contributions:**

This paper presents a method for interpolating between sentences in NLP, with the goal of enabling mixup data augmentation.  Four criteria are defined which interpolation models should exhibit. The paper then presents a model and training objective for placing a distribution over interpolated sentences. A set of hidden vectors is constructed by interpolating the lengths of the two sentences and interpolating between vectors that represent averages of words around the corresponding position in each sentence; e.g., the vector 50% of the way through the interpolated sentence will be an interpolation of a smoothed version of the vectors roughly 50% of the way through a and the vectors roughly 50% of the way through b. Training encourages reconstruction of a and b and uses regularization to improve performance. Results on several tasks show improvements from applying this method over baselines such as EDA and SSMBA.

The core claims of this paper (which are crucial for evaluating TMLR papers) are not super clear, but here are the two claims I will consider in my review (quoted from the paper's introduction):

Claim 1: "We manually inspect interpolated text and observe that LINDA is able to interpolate between any given pair of text with its strength controlled by the mixup ratio."

Claim 2: "LINDA outperforms existing data augmentation methods in the in-domain and achieves a competitive performance in OOD settings. LINDA also shows its strength in low-resource training settings."

**Audience:**

Yes

**Claims And Evidence:**

No

**Requested Changes:**

I believe if Claim 1 is better validated or toned down, this paper would be better positioned with respect to its claims and would be okay. This could involve better definition of interpolation and evaluation with respect to that definition (see above under "Weaknesses") or removal of the claim that the interpolation is really doing something meaningful, and instead backing off to the claim that this is simply some kind of useful data augmentation.

**Strengths And Weaknesses:**

STRENGTHS

This paper presents a nice, intuitively reasonable idea extending prior work on mixup for NLP. The solution to the problem of diverging lengths is clever and the model generally makes sense.  The four criteria defined for interpolation are a nice first step towards more formally defining how interpolation should work.

This paper itself is very clearly written. I was not familiar with all of these past approaches but I felt that I came away with a clear sense of where the state-of-the-art in this area lies.

The comparisons to prior methods and experimental evaluation are very thorough; several datasets across several different scenarios (low-resource, OOD, etc.) are examined.  Multiple methods from prior work are benchmarked in what appears to be an apples-to-apples comparison.

The low-resource results and NLI results are especially strong and compare quite nicely to prior work.

WEAKNESSES

For Claim 1, I am not all that convinced by the results presented in this work. Figure 2 shows that there is some success in getting lexical overlap and coarse semantic similarity to follow the expected trends when doing mixup. However, Table 1 seems to show poor results (generating weird sentences) when the sentences are dissimilar (first example, last example).  Table 8 in the appendix also seems like it starts with a pair where interpolation is not really defined and we do not see real interpolation happening as a result. As a result, I would judge Claim 1 to not be fully substantiated by the paper.  What I would like to see at a minimum would be (1) a clear definition of what interpolation should achieve empirically, beyond the theoretical criteria C1-C4. This probably requires restricting to a subset of sentence pairs where we can actually interpolate between two sentences in a meaningful way.  (2) Preferably human evaluation of whether interpolation is successful.

As for Claim 2, I think the results of this paper are mixed. As I said above, the low-resource and NLI results are quite strong. However, full performance in Table 2 doesn't show a clear trend between LINDA and LINDA_soft, and some of the gains are quite small and hard to know significance of (TREC-Coarse).  Nevertheless, LINDA and LINDA_soft both appear to have higher performance than the baselines when averaged across the datasets.

Out-of-domain task performance (Table 4) shows pretty minor and uneven gains, with basic LINDA decreasing performance on 3 of 4 datasets. LINDA_soft is a bit worse than SSMBA_soft. While this is appropriately described in the text, it does seem to undercut what's stated in the introduction.  As a result, I think Claim 2 is maybe 70% supported.

Taken together, I think ultimately this paper's claims are not quite accurately supported.

---

> ### Author Response · Authors · 2022-12-14
> **Response to Reviewer jTWk**
>
> We sincerely appreciate your efforts and insightful comments to improve the manuscript. In the revised manuscript, we have marked the revisions with “brown.”
>
> ## Meaning of Text Interpolation and Human evaluation
> There’s no consensus on the meaning of interpolation in the text domain in the NLP research field. Our work is one of the first works to attempt to define the text interpolation and we do so by laying out four conditions presented in the paper. Furthermore, we’d like to point out that text interpolation we discovered through LINDA is for data augmentation and no need to be aligned with what humans perceived as successful text interpolation. Also the ideal output of text interpolation when two sentences are given differs for every individual. However, we do agree that the suggested method is not the only method of defining interpolation in the text domain and that there could be other, possibly better, methods as well.
>
> ## Concern about Claim 1
>
> We agree with your concern and tone down our claim in revised manuscripts (section  5).

---

### Review · Reviewer_F22n · 2022-11-30

**Summary Of Contributions:**

This paper focuses on the topic of mixup in data augmentation. The key contribution is proposing an unsupervised learning approach to interpolate two sequences of different lengths and discrete tokens. In contrast to existing studies that relied on heuristics-based interpolation, the proposed method learns to interpolate between any pair of natural language sentences over a natural language manifold, which does not require any heuristics or manually crafted resources. Experimental results show that the proposed method (i) can interpolate two natural language sentences over a natural language manifold of well-formed sentences both quantitatively and qualitatively; (ii)  outperforms various data augmentation methods on various text classification datasets.

**Audience:**

Yes

**Broader Impact Concerns:**

The authors have provided a Broader Impact and Ethical Implications section, which is informative.

**Claims And Evidence:**

Yes

**Requested Changes:**

There are many presentation issues that should be fixed.

1. There are many references in incorrect formats. Please use \citet{} for inner-text citations.
"with domain shift in the test time Lim et al. (2019)" -> "with domain shift in the test time (Lim et al., 2019)"
"In the case of back-translation Edunov et al. (2018)"  -> "In the case of back-translation (Edunov et al., 2018)"
"including rule-based approaches Wei & Zou (2019b)"  -> "including rule-based approaches (Wei & Zou , 2019b)"

2. The presentation in Section 1 could be improved.

(i) The second paragraph discusses the two categories of data augmentation, which is hard to read. It is better to clearly indicate what the two categories are (e.g., input perturbation and back-translation) and use logical connectives to separate which part belongs to which category.

(ii) The motivation for the newly proposed method needs to be clarified. The third paragraph only talks about the issues of existing algorithms in the second category and how to overcome them. Why is the second category the focus of this paper? How is it more significant than the first one?

3. A few mathematical mistakes. In Section 3.4 Line 5, use $\tilde$ just on $x$ and $y$ (just leaving $m$).

**Strengths And Weaknesses:**

Strengths:

1. A new unsupervised method is proposed for applying mixup to variable-length discrete sequences without needing any heuristics or manually crafted resources.

2. The proposed technique is supposed to be generally effective (as an effective pluggable tool) for different backbone models (though the experiments on downstream tasks are just based on BERT-base).

3. The experimental results are comprehensive and convincing, with mean and standard deviations reported.

Weaknesses:

1. The proposed method (LINDA) requires massive computation for training compared with existing simple data augmentation methods like EDA. LINDA was trained on a million sentence pairs randomly drawn from English Wikipedia. When applying the proposed method to other languages, the training cost is indispensable.

2. Only one backbone model (BERT-base) is evaluated for downstream tasks. There is also a need for more clarification in selecting the backbone model. LINDA is trained by BART-large, which indicates that large models are affordable for the experiments. Why not select the large models as well for downstream tasks but use a relatively weaker BERT-base method instead?

3. Using LINDA does not achieve substantial performance gains over the vanilla baseline (Tables 2-3). It seems that LINDA is only better than the baseline by 0.1-0.8 (6/8) datasets. Taking the standard deviations into account, the results are mostly similar to the baseline. The authors implied that the proposed data augmentation methods are even inferior. Then, another issue is why we need those ``data augmentation" methods (including the proposed one) as they could not really effectively improve the performance.

4. In the low-resource setting, LINDA performs much better than the vanilla method in terms of mean accuracies. There are two concerns in the comparison. The first is when we consider the standard deviations, there is a large overlap between the accuracies of LINDA and the baseline. It still needs to be made clear whether LINDA is indeed significantly better than the baseline. I would recommend a statistical test. The second concern is that the comparison might not be fair. The baseline is only trained with only 5 or 10 randomly selected examples per class, but LINDA is trained on a million sentence pairs.

---

> ### Author Response · Authors · 2022-12-14
> **Response to Reviewer F22n**
>
> We sincerely appreciate your efforts and insightful comments to improve the manuscript (section1). In the revised manuscript, we have marked the revisions with “brown.”
>
> ## Training Cost for LINDA
> The core motivation of our work is 1) to define text interpolation and its conditions to be met  2) to make the text interpolation process learnable and the training process comes naturally as we are suggesting a method to learn to interpolate following definitions. We’d like to emphasize that although LINDA requires training on a million sentence pairs of Wikipedia, LINDA does not require any further fine-tuning when applied to data augmentation tasks on downstream datasets once it’s trained. Whereas existing data augmentation methods such as [1, 2]] require fine-tuning on the downstream datasets as they perform augmentation.
>
> ## Choice of BART-large for LINDA and BERT-base for Downstream Tasks
> Since there is no consensus on how to use MLM models such as BERT as a generator, we choose BART-large, which has the ability to generate, as the backbone architecture of LINDA. During experiments on downstream datasets, our main focus was to show a fair comparison using the same classifier. Since  various data augmentation works [3,4] use BERT-base for sentence classification tasks, we follow these experiment settings for  downstream task classifiers.
>
>
> [1] He, Xuanli et al. “Generate, Annotate, and Learn: NLP with Synthetic Text.” Transactions of the Association for Computational Linguistics 10 (2021): 826-842.[2] Wu, Xing et al. “Conditional BERT Contextual Augmentation.” ArXiv abs/1812.06705 (2018): n. pag.
> [3] Yoon, Soyoung et al. “SSMix: Saliency-Based Span Mixup for Text Classification.” ArXiv abs/2106.08062 (2021): n. pag.
> [4]Kumar, Varun et al. “Data Augmentation using Pre-trained Transformer Models.” ArXiv abs/2003.02245 (2020): n. pag.

---

### Review · Reviewer_gTMr · 2022-12-06

**Summary Of Contributions:**

The paper presents a new method for mix-up data augmentation and evaluates it on single sentence and pairwise classification tasks in low-resource, standard, and domain adaptation settings. The method includes an approach to train a sequence-to-sequence model, which generates synthetic text given two input text sequences and an interpolation parameter \alpha. Since there is no annotated training data for this augmentation task, the paper  uses regularization, constraints and tailored computation of activations to derive a model that is shown to have the desired properties of being closer to the first or second input depending on \alpha while maintaining fluency.

The method is shown to be comparable to outperform strong baselines in a variety of settings.

**Audience:**

Yes

**Broader Impact Concerns:**

No concerns

**Claims And Evidence:**

Yes

**Requested Changes:**

I have some questions that can be clarified in a revision:

*  For the baseline SSMBA, which masked language model is used and is it supposed to be as good as the BART-large starting model in your approach?

* For pairwise classification tasks, was there an impact of whether augmentation is done for the hypothesis/premise parts jointly versus independently? If there is an effect, was the better setting also used for the baseline methods (e.g. SSMBA)

* Were the regularizers ablated?



Minor: The formatting of the references is not right at multiple places, e.g. BART Lewis et al. (2020) → BART (Lewis et al. 2020)


**Strengths And Weaknesses:**

Strengths
 * The paper is written well
 * The motivation is good, and the method is novel and creative
 * The method shows advantages over strong well chosen baselines
 * The experiments and analysis are thorough

Weaknesses
 * Evaluation is limited to text and text-pair classification tasks

---

> ### Author Response · Authors · 2022-12-14
> **Response to Reviewer gTMr**
>
> We sincerely appreciate your efforts and insightful comments to improve the manuscript. In the revised manuscript, we have marked the revisions with “brown.”
>
> ### Clarification with SSMBA
>
> We used BERT-base for SSMBA with the same setting as in paper. As BART model is based on an auto-encoder architecture different from BERT’s, it is hard to directly replace the BERT to BART model. However, we suggest similar works [1] such as adjusting token or span masking to data and reconstructing the augmented data using denoising autoencoders for only on low-resource settings.
>
> ### Experimental Setting of NLI
> We appreciate the reviewer’s suggestion. Since LINDA is a single sentence-level data augmentation method, it was natural to mix a hypothesis with another hypothesis and a premise with another premise. We agree that applying single sentence-level augmentation models to multi-sentence settings is an important research topic..
>
> ### Ablation on Regularizers
> We have empirically checked that LINDA performs better by adding each regularizer. However, since our paper doesn’t report the downstream task performance difference regarding the application of regularizers, we’d like to explain further how each regularization helps the training of LINDA.
> These regularizers are however pretty much standard in training a language model, and we have already described justification for each of these in the text. More specifically, masking discourages carbon copies of tokens (satisfying Condition 4), and noise in the hidden representation together with norm regularization ensures the hidden space is densely populated, which facilitates linear interpolation in the hidden space.
>
> [1] Kumar, Varun et al. “Data Augmentation using Pre-trained Transformer Models.” ArXiv abs/2003.02245 (2020): n. pag.

---

### Author Response · Authors · 2022-12-14
**General Response**

Dear reviewers and Editors,

We sincerely appreciate your valuable time and effort spent reviewing our manuscript. As reviewers highlighted, our aims at an important and well-motivated (reviewer gTMr) problem, and provide a novel and creative (all reviewers), yet intuitive (jTWk) method that shows strong empirical results (all reviewers) on the thoroughly extensive experiments (all reviewers) with clear writing (gTMr, JTWk). In particular, we believe that LINDA makes a meaningful contribution to exploring a definition of text interpolation, as highlighted by reviewer gTMr and jTWk.

We appreciate your constructive feedback on our manuscript. In response to the comments, we have carefully revised the manuscript, including in-correct citation and equation, clear explanation about related work and motivation (Section 1) and tone down our claims (section 5).
In the revised manuscript, these updates are temporarily highlighted in "brown” for your convenience to check.

Moreover, we would like to respond to common concerns about performance.

Full Setting: Reporting experimental results of LINDA on various settings (full, low resource, OOD) was to show the behavior of data augmentation methods, including LINDA, under diverse circumstances. We’d like to point out that our intention to report full dataset performance was not to argue that LINDA gives significant performance gain under full dataset setting but to compare which circumstance data augmentation works the best.

Low Resource Setting: For low resource setting experiments, since we randomly sample 5 / 10 data examples for each run, high variance is inevitable. It will be important to select a model based on the average performance in practice.

Although we agree with the concern that we report only minor improvements over other augmentation methods, we want to emphasize learning to interpolate is a novel conceptual framework and has a potential for further improvement in follow-up studies, with e.g. better parametrization, which we will leave for the future.

Thank you very much,

Authors.

---

### Decision · Action_Editors · 2023-01-10

**Recommendation:** Reject

**Comment:**

See above comments.

**Audience:**

Yes

**Claims And Evidence:**

I like the problem studied in the paper. I share the same concerns as Reviewer jTWk: the two claims are not well supported.

First, the definition of text interpolation is unclear, and the interpolation capability of the proposed method is not well justified, no matter theoretically/analytically or empirically.

-  The four conditions C1-C4 are a bit trivial and they might not be essential to text interpolation. Given that the main challenges of mixup for NLP come from that two sequences are of different lengths and discrete tokens, can we do some analysis on the implications of the different lengths and discrete tokens? Or can we define some conditions or constraints on these two factors? I highly agree that "there’s no consensus on the meaning of interpolation in the text domain in the NLP research field"; actually considering this situation, it is more demanding and impactful to do some deeper analysis to approach consensus.
- For Table 1, it is better to show the mixup results with different ratio $\alpha$'s for a fixed pair of input sequences, which will provide more insights to the community.

Second, empirical improvements look marginal, as pointed out by reviewers.

Overall, the work has the potential to be a high-quality paper, but its current version needs significant efforts to address the concerns raised by reviewers.